# Pentagone internalises glypicans to fine-tune multiple signalling pathways

**Mark Norman[1], Robin Vuilleumier[2], Alexander Springhorn[2,3], Jennifer Gawlik[1,3], George Pyrowolakis[1,2]***

[1]Centre for Biological Signalling Studies, Albert-Ludwigs-University of Freiburg, Breisgau, Germany; [2]Institute for Biology I, Albert-Ludwigs-University of Freiburg, Breisgau, Germany; [3]Spemann Graduate School of Biology and Medicine, Albert-Ludwigs-University of Freiburg, Breisgau, Germany

**Abstract** Tight regulation of signalling activity is crucial for proper tissue patterning and growth. Here we investigate the function of Pentagone (Pent), a secreted protein that acts in a regulatory feedback during establishment and maintenance of BMP/Dpp morphogen signalling during Drosophila wing development. We show that Pent internalises the Dpp co-receptors, the glypicans Dally and Dally-like protein (Dlp), and propose that this internalisation is important in the establishment of a long range Dpp gradient. Pent-induced endocytosis and degradation of glypicans requires dynamin- and Rab5, but not clathrin or active BMP signalling. Thus, Pent modifies the ability of cells to trap and transduce BMP by fine-tuning the levels of the BMP reception system at the plasma membrane. In addition, and in accordance with the role of glypicans in multiple signalling pathways, we establish a requirement of Pent for Wg signalling. Our data propose a novel mechanism by which morphogen signalling is regulated.

***For correspondence:** g.pyrowolakis@biologie.uni-freiburg.de

**Competing interests:** The authors declare that no competing interests exist.

## Introduction

Bone morphogenetic protein (BMP) signalling is required in a wide variety of processes across higher organisms, from the establishment of the dorso-ventral (DV) axis in insects to the maintenance of the mammalian gut (*Ferguson and Anderson, 1992*; *Haramis et al., 2004*). Many of the biological functions of BMP signalling require a high degree of spatial regulation; accordingly, multiple mechanisms have evolved that control the movement, stability and activity of BMP ligands (*Brazil et al., 2015*; *Ramel and Hill, 2012*; *Umulis et al., 2009*; *Wharton and Serpe, 2013*).

One of the most intensely studied examples of extracellular regulation of BMP comes from *Drosophila* wing development, a tissue where Dpp (Drosophila BMP2/4) acts as a morphogen to control both patterning and growth (*Restrepo et al., 2014*; *Wartlick et al., 2011*). During larval wing development, Dpp is produced in a stripe of cells at the anterior-posterior (AP) boundary and disperses into both compartments, by mechanisms that are still not fully understood, to organize a BMP signalling activity gradient along the AP axis with highest levels in medial and lowest in lateral regions (*Affolter and Basler, 2007*). Dpp, together with a second, uniformly expressed ligand, Glass bottom boat (Gbb), activates membrane bound receptors and induces the phosphorylation of the transcription factor Mad. Phosphorylated Mad (pMad) accumulates in the nucleus with the cofactor Medea, where the activated Smad complex directly regulates BMP-target gene transcription (*Hamaratoglu et al., 2014*).

In addition to the localized production of Dpp, many other determinants impact on proper establishment and maintenance of the activity gradient. Prominent amongst them are membrane-bound BMP-binding proteins, such as Thickveins (Tkv) and Dally, which have dual functions in the establishment of the pMad gradient. Tkv is cell-autonomously required for signalling as it is the main type I

BMP receptor in *Drosophila*. At the same time, Tkv critically affects Dpp tissue distribution through ligand trapping and internalisation, and thus globally shapes the BMP activity gradient (*Crickmore and Mann, 2006*; *Lecuit and Cohen, 1998*; *Tanimoto et al., 2000*). Similarly, Dally, a GPI-anchored heparan sulfate proteoglycan (HSPG), binds and concentrates Dpp at cell surfaces and, together with a second glypican, Dally-like protein (Dlp), is required for both local signal activation and long-range distribution of the ligand (*Akiyama et al., 2008*; *Belenkaya et al., 2004*). Absence of glypicans- for example in a clone of cells- can result in both a reduction of BMP signalling within the clone and an interruption of Dpp spreading within and beyond the clone. In addition, glypicans can also, by virtue of their ligand-binding capacity, hinder the movement of ligands in several contexts. For example, the diffusion of BMP4 (a vertebrate homolog to Dpp) during early Xenopus embryogenesis has been shown to be restricted through its interactions with HSPGs (*Ohkawara et al., 2002*). Similarly, in the wing disc, increasing the levels of Dally at the source of Dpp causes a local increase of signalling activity and a drastic compaction of the gradient due to ligand trapping (*Fujise et al., 2003*; and see below). Reflecting the importance of the activity of Tkv and Dally for proper gradient establishment, the levels of both proteins are tightly regulated along the AP axis of the developing disc. Through complex transcriptional regulation, which involves repression by BMP signalling itself, both Tkv and Dally are down-regulated near the ligand source to maintain the proper balance between Dpp signalling and Dpp dispersion (*Crickmore and Mann, 2006*; *Lecuit and Cohen, 1998*; *Tanimoto et al., 2000*; *Fujise et al., 2001*).

We have previously described Pentagone (Pent; also known as Magu), an additional determinant in the establishment, maintenance and scaling of the BMP signalling gradient in the developing wing (*Ben-Zvi et al., 2011*; *Hamaratoglu et al., 2011*; *Li and Tower, 2009*; *Vuilleumier et al., 2010*; *Zheng et al., 2011*). The transcription of *pent* is directly repressed by BMP signalling, hence its production is restricted to the lateral-most cells of the disc. Pent protein is, however, secreted and distributes in a gradient that is inverse to the pMad gradient. Pent mutants have a restricted pMad gradient with abnormally high levels in the centre of the disc and very low levels in lateral regions; consequently, adult wings have growth and patterning defects in lateral regions. The pMad gradient of *pent* mutants thus resembles the abnormal gradients caused by medial over-expression of Tkv or Dally, suggesting an interaction of the protein with the BMP-reception system. Indeed, our past work established that Pent physically associates with Dally on cell membranes, but the consequences of this interaction have remained unclear.

In this study we present data showing that Pent binds and induces the internalisation of both *Drosophila* glypicans, resulting in reduction of Dally and Dlp protein levels. Endocytosis of glypicans is dependent on dynamin and Rab5, but does not require clathrin or Dpp signalling. Additionally, we show that Pent influences Wg signalling, which also depends on glypicans. We conclude from these data that Pent modulates glypican levels in order to modify multiple signalling pathways during wing morphogenesis. Our data suggest an additional, protein-level feedback mechanism to tightly control levels of signalling, which cooperates with transcriptional regulatory feedback loops to ensure proper morphogen gradient formation and organ development.

## Results

### Pent induces internalisation of glypicans

Glypicans have been demonstrated to be critical for the correct formation of the Dpp signalling gradient in the wing imaginal disc. While binding of Dpp to glypicans can promote signalling and movement of Dpp, glypicans can also block ligand dispersion. Increasing the level of Dally in the centre of the disc by dppGal4 increases pMad and Dpp levels medially in a cell-autonomous manner (*Fujise et al., 2003*), but this is at the expense of lateral signalling as judged by the drastic restriction of the pMad gradient (*Figure 1—figure supplement 1A–C*). Assessing the distribution of Dpp itself by an extracellular staining protocol reveals accumulation of the ligand on Dally-expressing cells and a simultaneous loss of Dpp in cells flanking the source, demonstrating that too much Dally can trap Dpp and impede its spreading (*Figure 1—figure supplement 1D–F*).

We have previously shown that Pent physically interacts with the glypican Dally and that the heparan sulphate side chains of the glypican are required for Pent binding (*Vuilleumier et al., 2010*) (*Figure 1—figure supplement 2A–D,G,H*). These experiments suggest that Pent can be found on

glypicans far from the source of Pent production. To better investigate the physiological distribution of Pent we used genomic engineering to generate flies expressing a N-terminally YFP tagged Pent (PentYFP) under endogenous control (see Materials and methods). Flies with PentYFP as the only source of Pent display normal wing morphology and a normal pMad gradient, demonstrating that the fusion protein is fully functional (*Figure 1—figure supplement 3A,E,F,A',E',F',I*). In such flies, PentYFP was mostly detectable in lateral regions of the wing imaginal discs (*Figure 1—figure supplement 2K*) and was below detection levels in medial cells. However, when we expressed an anti-GFP nanotrap - a membrane-tethered, extracellularly exposed single-chain nanobody against GFP (*Harmansa et al., 2015*) - using dppGal4, PentYFP strongly accumulated on nanotrap expressing cells in the centre of the disc (*Figure 1—figure supplement 2K,L*). This clearly demonstrates the presence of extracellular Pent in the centre of the wing disc, far from its production source.

To address the molecular function of Pent, we first sought to understand the consequences of the interaction between Dally and Pent. We used a DallyYFP fusion protein, where YFP is inserted as an artificial exon between the first and second exons of the gene, to monitor Dally levels and distribution (*Lowe et al., 2014*). This chimeric protein is functional as it can restore the adult wing defects and the abnormal pMad profile observed in *dally* mutants (*Figure 1—figure supplement 3A–C,A–C',I'*). In the third instar wing disc, DallyYFP protein localised at cell membranes and, consistent with previous reports, was found at its highest levels in the lateral regions and lowest in the medial region of the disc (*Figure 1A*) (*Ayers et al., 2012*; *Fujise et al., 2001*). Increased levels of DallyYFP were also seen at the DV and AP boundaries, although the latter could notalways be detected. To investigate the effect of Pent on Dally, we expressed V5Pent in the dorsal compartment of the wing disc using apterous-Gal4 (apGal4). As the profile of Dally is mirrored in the DV axis, the ventral compartment served as an internal control in these experiments. The distribution of DallyYFP was dramatically altered upon expression of V5Pent (*Figure 1B*), with a marked redistribution of the protein from membranes to discrete puncta. Closer inspection showed that V5Pent and DallyYFP co-localised in these puncta, with 66% of V5Pent puncta found to also contain DallyYFP (*Figure 1B,C*). The majority of V5Pent-DallyYFP puncta were found in the dorsal compartment, where expression of V5Pent was driven by apGal4 (*Figure 1B*). Nevertheless, and consistent with the spreading of Pent, co-localisation of Pent and DallyYFP in puncta was also seen in ventral cells multiple cell-diameters away from the DV boundary (*Figure 1B*). Clonal expression of V5Pent also resulted in redistribution of DallyYFP and the appearance of V5Pent-DallyYFP puncta (data not shown).

To directly test whether the appearance of puncta is at the expense of the membrane-anchored pool of Dally, we used an extracellular staining protocol with a fluorescently labelled anti-GFP nanobody (*Strigini and Cohen, 2000*). This assay revealed a substantial reduction of extracellular Dally upon V5Pent expression (*Figure 1D*). The reduction in DallyYFP protein level is not due to a reduction in Dally mRNA as a reporter of Dally transcription showed no significant reduction upon UASV5-Pent expression (data not shown). Importantly, DallyYFP and Pent also co-localise in puncta at endogenous protein levels, although at lower frequency than that seen in over-expression studies (*Figure 1—figure supplement 4A*).

To characterize the DallyYFP and V5Pent puncta, we co-stained for the endosomal markers Hrs, Rab7 and Rab11 (*Figure 1E* and *Figure 1—figure supplement 4B,C*). Frequent co-localisation was observed with both Rab7 and Hrs, showing that these DallyYFP-V5Pent puncta were in endosomal compartments (*Figure 1E* and *Figure 1—figure supplement 4C*). In particular, most V5Pent-DallyYFP endosomes were positive for the late endosomal marker Rab7 (*Figure 1C*), indicating that they are on the route to the lysosome and degradation. No co-localisation was seen with Rab11, a marker for recycling endosomes, indicating that DallyYFP-V5Pent endosomes are not recycled back to the plasma membrane (*Figure 1—figure supplement 4B*). The localisation of V5Pent and DallyYFP to late but not recycling endosomes correlates with the drastic reduction in DallyYFP protein level observed.

To address whether Dally protein levels are enhanced in the absence of Pent, we monitored the fluorescent intensity of DallyYFP in wild type and *pent* mutant discs. In comparison to wild type discs, a small increase in DallyYFP was observed in the anterior compartment of *pent* discs, and a more substantial increase in the posterior compartment (*Figure 1F,G*). The pattern of Dally distribution was also notably changed, from a smooth gradient in wild type discs to a much more abrupt gradient in *pent* discs, with a sharp increase in the posterior compartment. Importantly, we saw a similar increase in the level of extracellular Dally in *pent* mutant discs (*Figure 1G*). Transcriptional

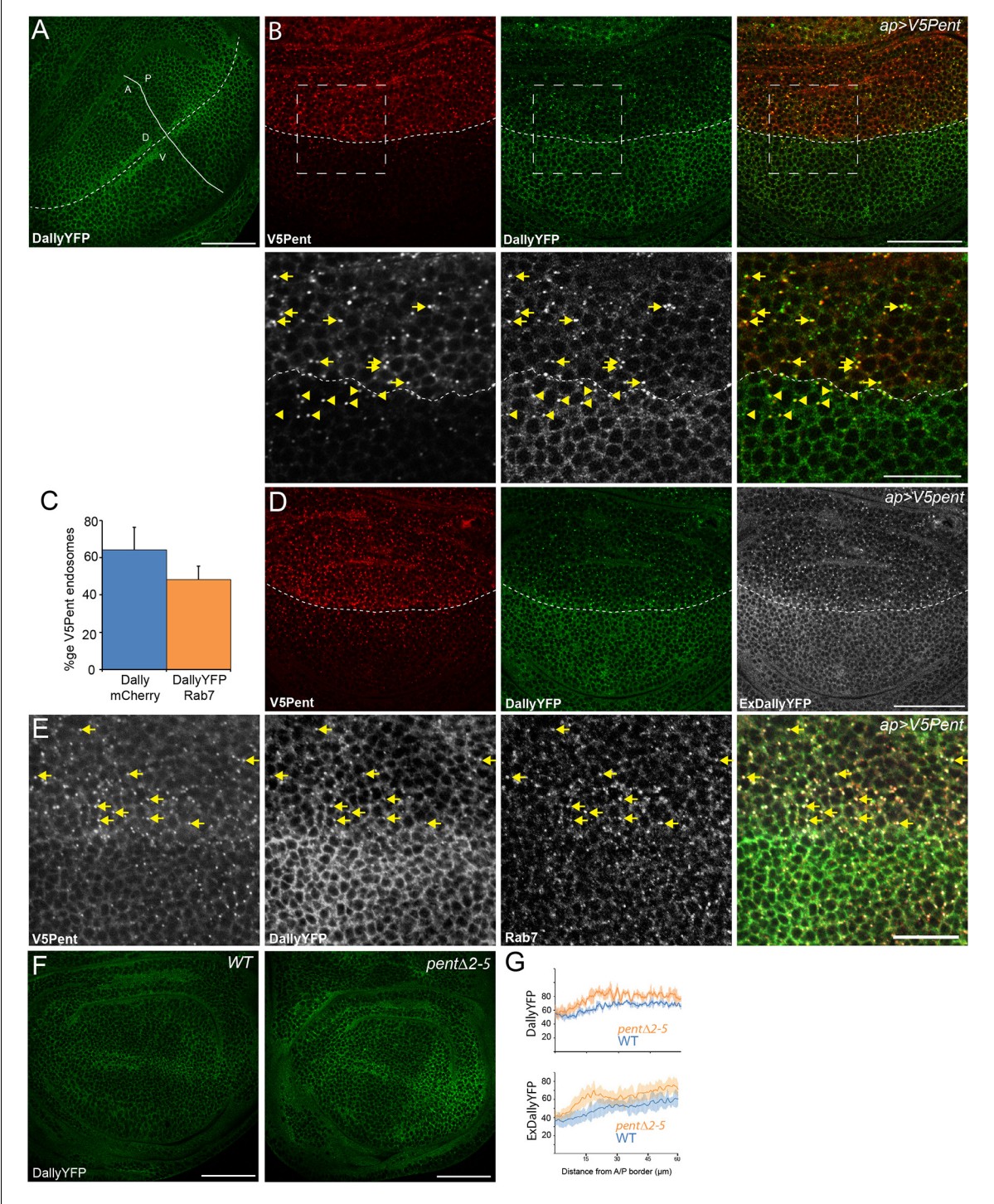

**Figure 1.** Pent internalises Dally. (**A**) Localisation of DallyYFP in a WT third instar wing disc. Solid and dashed lines indicate the AP and DV boundaries respectively. (**B**) Expression of UASV5Pent with apGal4 internalised DallyYFP into puncta which co-localise with V5Pent. Arrows and arrowheads show co-localisation in the dorsal compartment and ventral compartments respectively. Dashed line shows DV boundary. (**C**) Quantification of V5Pent co-localisation with a functional DallymCherry fusion protein (for details on this line see Materials and methods and *Figure 1—figure supplement 3A, D, A', D', I*) and Rab7YFP in discs expressing UASV5Pent driven by apGal4. 66% of V5Pent endosomes contained Dally, and 49% contained Dally and Rab7. Error bar represents standard deviation (n=10). (**D**) Expression of UASV5Pent with apgal4 reduces extracellular DallyYFP (grey, right panel) as well as total DallyYFP (green, left). (**E**) DallyYFP and V5Pent co-localise with the late endosomal marker Rab7. Disc expressing UASV5Pent with apGal4. Endosomes positive for DallyYFP, V5Pent and Rab7 are marked with arrows. (**F, G**) DallyYFP protein level is increased in *pent* mutant wing discs. Images show third instar wing discs with DallyYFP autofluorescence in *WT* and *pentΔ2–5*. The graph shows mean data of quantification of fluorescence
*Figure 1 continued on next page*

*Figure 1 continued*

intensity and extracellular labelling of DallyYFP in the posterior compartment (n=30 in three independent experiments, error bars show standard deviation). Scale bars are 50 μm in upper images and 20 μm in zooms and (E). See also *Figure 1—figure supplements 1–4*.

The following figure supplements are available for figure 1:

**Figure supplement 1.** Excessive Dally traps Dpp.

**Figure supplement 2.** Binding interactions and distribution of Pent.

**Figure supplement 3.** Endogenously tagged BMP signalling components are fully functional.

**Figure supplement 4.** Pent and DallyYFP co-localise at endogenous levels.

regulation of *dally* expression by the altered pMad gradient in *pent* mutants certainly contributes to the observed changes, especially the medial expansion of DallyYFP expression. Nevertheless, the increase of DallyYFP in lateral-most cells (where pMad signalling is absent) is likely due to direct effects of Pent on Dally stability and is consistent with our gain of function analysis. Cumulatively, the data suggest that Pent induces the internalization of Dally and its depletion from the plasma membrane.

To test whether the effects of Pent are restricted to Dally or generally applicable to glypicans, we monitored the interaction with the second *Drosophila* glypican, Dlp. Pent accumulates on cells over-expressing Dlp-GFP, suggesting that Pent is able to bind Dlp as well as Dally (*Figure 1—figure supplement 2E,G,I*); we therefore asked whether Pent is also able to internalise Dlp. In wild type conditions, Dlp is found at highest levels either side of the DV boundary, with levels low at the DV boundary itself (*Figure 2A*). We expressed V5Pent in the posterior compartment of discs as this allows easy comparison of expressing and non-expressing cells (*Figure 2B*). Similar to what was seen with DallyYFP, the level of Dlp protein was drastically reduced in the posterior compartment. The remaining Dlp protein was localised to puncta that frequently co-localised with V5Pent, with 57% of V5Pent puncta also containing Dlp (*Figure 2B,C*). These puncta were also destined for lysosomal degradation as they were positive for the late endosome marker Rab7 (*Figure 2C,D*).

To monitor the level of Dlp in *pent* mutant discs, we used a functional DlpYFP fusion protein (*Figure 1—figure supplement 3G,G',I*). As with Dally, DlpYFP levels were slightly increased across *pent* mutant discs in comparison to WT control (*Figure 2E,F*), a result also confirmed by immunoblotting (*Figure 2G*). Unlike Dally, the pattern of Dlp transcription does not suggest it is suppressed by Dpp signalling. Indeed, we could not detect changes in Dlp protein level in *mad* mutant clones, which have no Dpp signalling (*Figure 2—figure supplement 1*). Therefore, the increase in DlpYFP seen in *pent* mutants is likely due to reduced internalisation of Dlp by Pent. These data show that Pent influences glypicans in general, and not just Dally. Glypican levels are reduced when Pent is over-expressed, and their levels increase in the absence of Pent. Consistent with the role of glypicans in Dpp signalling and the phenotypes observed in glypican loss of function studies, the reduction of glypicans by Pent over-expression results in aberrant BMP signalling and gradient formation. In discs expressing Pent in the dorsal compartment we saw a local decrease in extracellular Dpp levels and a concomitant shrinkage of the pMad gradient (*Figure 2—figure supplement 2*).

In addition to Pent, glypicans are able to bind many proteins crucial for wing development, including Dpp, the Wg-regulator Notum, and Shifted (Shf), a regulator of Hh signalling (*Akiyama et al., 2008*; *Bilioni et al., 2013*; *Gerlitz and Basler, 2002*; *Giráldez et al., 2002*). To determine whether association of any binding protein can induce glypican endocytosis, we expressed Dpp, Notum and Shifted with apGal4 and monitored the level of DallyYFP. Notum and Dpp occasionally co-localised with Dally in endosomes, but neither Dpp, Notum nor Shf induced internalisation and depletion of Dally (*Figure 2—figure supplement 3A–C*). This leads us to conclude that the recruitment of glypicans into endosomes and consequent reduction in glypican protein level is a specific property of Pent and not a general consequence that occurs upon binding of an extracellular protein.

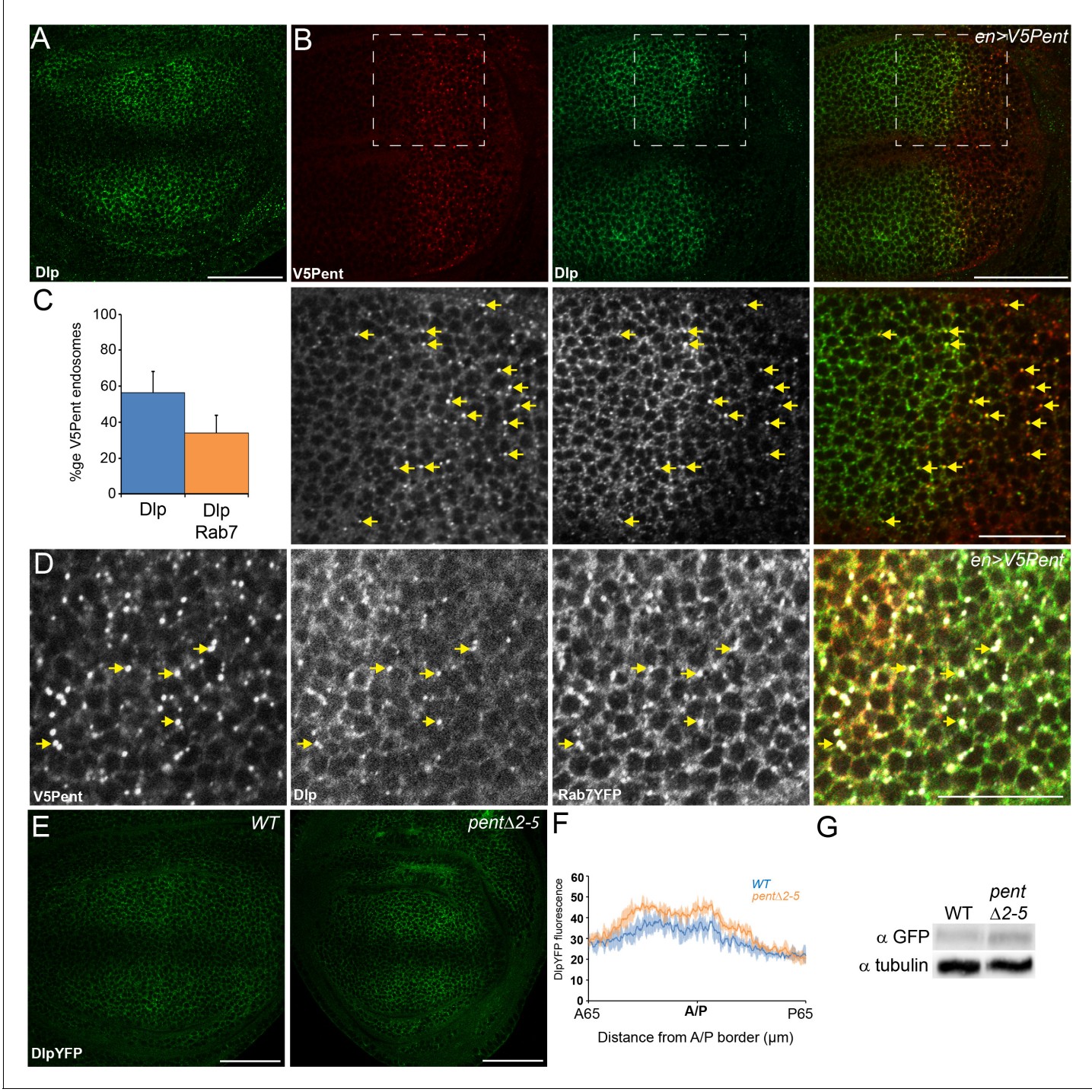

**Figure 2.** Pent internalises the second *D. melanogaster* glypican, Dally-like protein. (A) Localisation of Dlp in a WT third instar wing disc. (B) Expression of UASV5Pent with enGal4 resulted in a decrease of Dlp protein in the posterior compartment and co-localisation of Dlp with V5Pent. Arrows show endosomes where Dlp and V5Pent co-localise. (C) Quantification of V5Pent co-localisation with Dlp and Rab7YFP in discs expressing UASV5Pent driven by apGal4. 57% of V5Pent endosomes contained Dlp, and 34% contained Dlp and Rab7. Error bar represents standard deviation (n=10). (D) Dlp and V5Pent co-localise with the late endosomal marker Rab7. Disc expressing UASV5Pent with enGal4, in a Rab7YFP background. Endosomes positive for Dlp, V5Pent and Rab7 are marked with arrows. (E, F) DlpYFP protein level is increased in *pent* mutant wing discs. Images show third instar wing discs with DlpYFP autofluorescence in *WT* and *pentΔ2–5*. The graph shows mean data of quantification of fluorescence intensity (n=20 in three independent experiments, error bars show standard deviation). (G) Western blot showing an increase in DlpYFP in *pentΔ2–5* mutant discs compated to WT. Boxes are enlarged in the lower panels. Scale bars are 50 µm, except lower panels of (B) where they are 20 µm. See also *Figure 2—figure supplement 1–3*.
*Figure 2 continued on next page*

*Figure 2 continued*

The following figure supplements are available for figure 2:

**Figure supplement 1.** Dlp expression is not repressed by Dpp signalling.

**Figure supplement 2.** Pent in the dorsal compartment reduces Dpp accumulation and affects pMad gradient formation.

**Figure supplement 3.** Dpp, Notum and Shf do not internalise DallyYFP.

## Pent does not induce significant internalisation of Dpp receptors

After addressing the effect of Pent on the Dpp co-receptor Dally, we wished to examine potential interactions and effects on the Dpp receptor itself. We focused on Tkv and Punt, which account for most of the BMP signalling during wing disc development (*Ruberte et al., 1995*). Unlike Dally and Dlp, expression of Tkv in the dorsal compartment of the disc did not affect the distribution of Pent, suggesting that Pent does not bind Tkv (*Figure 1—figure supplement 2F,J*). To monitor effects of Pent on the level of Tkv protein, we generated an endogenous TkvHA fusion protein (see Materials and methods). Flies with TkvHA as the sole source of Tkv are fully viable and display no molecular or morphological defects in wing development, confirming the functionality of the fusion protein (*Figure 1—figure supplement 3H,H',I*). Under normal conditions, and consistent with previous studies, TkvHA is high laterally and low in the medial region with particular drops in protein level at the AP and DV boundaries (*Figure 3A*) (*Crickmore and Mann, 2006*; *Lecuit and Cohen, 1998*; *Tanimoto et al., 2000*). Unlike the dramatic effects seen with glypicans, TkvHA was not visibly reduced when Pent was over-expressed in the dorsal compartment (*Figure 3B*). However, similar to the effects seen with DallyYFP, TkvHA co-localised with V5Pent puncta (*Figure 3B*), which were frequently positive for the late endosome marker Rab7 (*Figure 3C,D*), suggesting these endosomes are destined for lysosomal degradation. As Pent is internalised with Tkv and Dally, we sought to determine whether Tkv and Dally co-localise into endosomes together. When V5Pent was expressed with apGal4, V5Pent-Dally-Tkv endosomes were frequently seen, showing that Dally and Tkv co-localise in endosomal compartments (*Figure 3—figure supplement 1A*). From these data we conclude that Pent may be able to drag Tkv into endosomes with Dally, but this happens at a frequency too low to visibly reduce Tkv protein level. Alternatively, the appearance of Dally and Tkv in the same endosomes could be due to downstream fusion of independent endosomes containing Tkv or Dally. In either scenario, Pent primarily exerts its function through regulation of glypicans, and not Tkv.

To examine potential interactions of Pent with the Dpp type II receptor, we expressed V5Pent with apGal4 and monitored the level of Punt using a Punt-GFP fusion protein encoded by a genomic rescue construct. There was no reduction in Punt protein level in the dorsal compartment and, unlike Tkv, there was also no co-localisation of Punt and Pent (*Figure 3—figure supplement 1B*). As Tkv-Punt dimerization occurs upon ligand binding, we hypothesize that Pent is primarily endocytosed with inactive Tkv receptors.

## Internalisation of glypicans by Pent is signalling independent

Lack of Punt internalisation by Pent suggests that active signalling is not required for Pent induced effects on glypican distribution. We used two additional approaches to directly examine whether internalisation of glypicans by Pent requires active Dpp signalling. Firstly, we clonally expressed V5Pent in discs expressing either Punt or Tkv RNAi under control of apGal4. Under these conditions, Dpp signalling is blocked in the dorsal compartment and is normal in the ventral compartment. DallyYFP was still internalised with V5Pent in the compartment expressing Tkv or Punt RNAi, and the level of DallyYFP was visibly reduced (*Figure 3—figure supplement 1C,D*). This demonstrates that receptors and active signalling are not required for Pent-induced glypican internalisation. In a further experiment, we sought to address whether Dpp itself, which also directly binds to glypicans, is required for Pent to internalise glypicans. Dpp is absolutely required for wing disc development; however, some wing discs do grow in *brk/dpp* double mutant flies (genotype $brk^{XA}$;$dpp^{d12}$/$dpp^{d14}$)

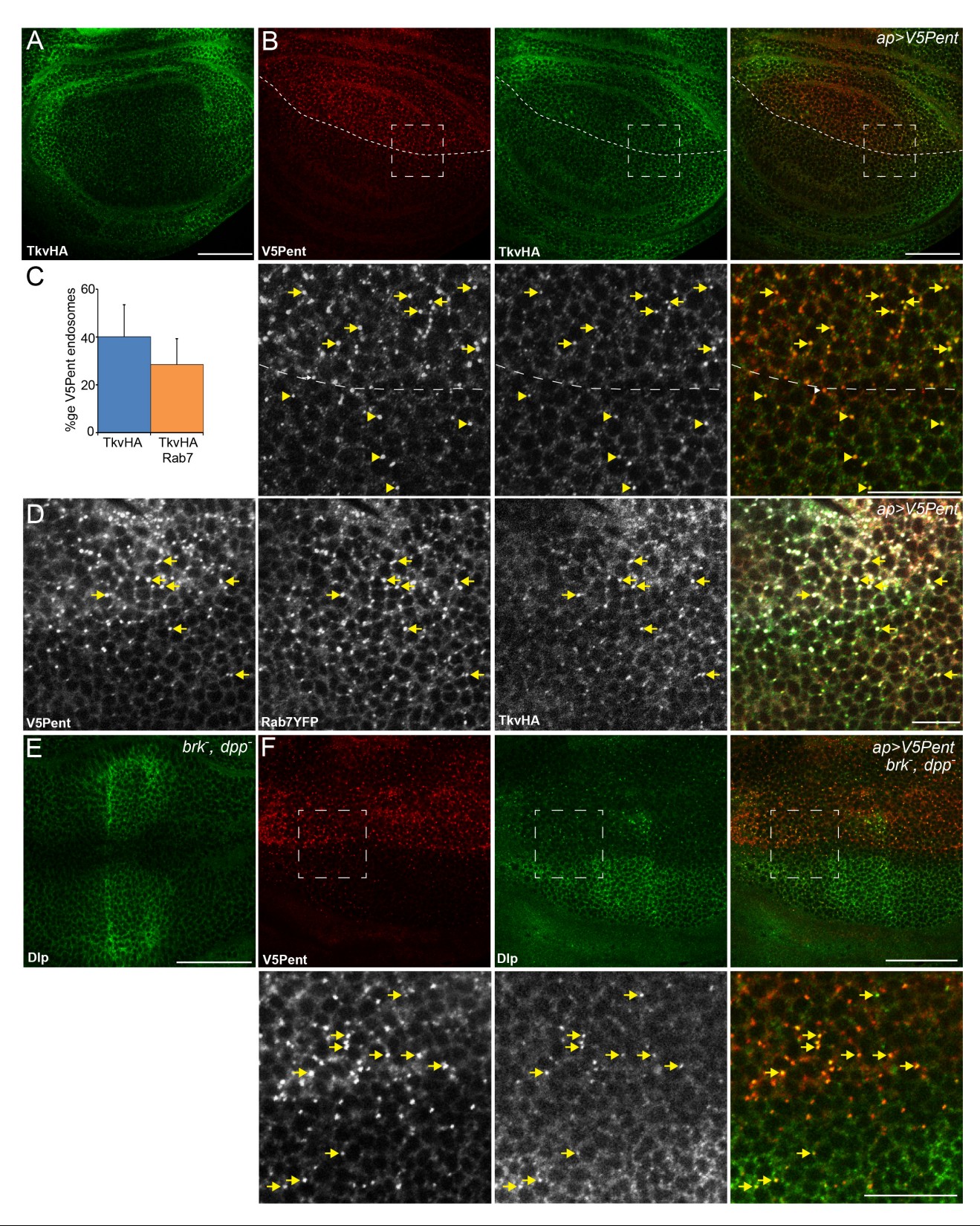

**Figure 3.** Internalisation of glypicans by Pent is signalling independent. (**A**) Localisation of TkvHA in a WT third instar wing disc. (**B**) UASV5Pent expressed with apGal4 co-localised with TkvHA in puncta, but no reduction in TkvHA protein level was seen. Arrows show co-localisation of Pent and
*Figure 3 continued on next page*

*Figure 3 continued*

TkvHA in the dorsal compartment, arrowheads in the ventral compartment. (C) Quantification of V5Pent co-localisation with TkvHA and Rab7YFP in discs expressing UASV5Pent driven by apGal4. 40% of V5Pent endosomes contained TkvHA, and 29% contained TkvHA and Rab7. Error bar represents standard deviation (n=10). (D) TkvHA and V5Pent co-localise with the late endosomal marker Rab7. Disc expressing UASV5Pent with apGal4. Endosomes positive for TkvHA, V5Pent and Rab7 are marked with arrows. (E) Localisation of Dlp in a *brk^XA*; *dpp^d12*/*dpp^d14* third instar wing disc. There is an increase in intensity in the centre of the disc. (F) Internalisation of Dlp by Pent does not require Dpp. Wing disc expressing UASV5Pent with apgal4 in a *dpp* mutant background. Arrows show co-localisation of V5Pent and Dlp in endosomes. Boxes indicate regions enlarged in lower panels. Scale bars represent 50 μm in normal images and 20 μm in enlarged regions. *Figure 3—figure supplement 1*.

The following figure supplement is available for figure 3:

**Figure supplement 1.** Internalisation of Dally by Pent does not require the receptor.

---

(*Campbell and Tomlinson, 1999*; *Schwank et al., 2008*). In the absence of Dpp, the distribution of Dlp was slightly different, with increased levels at the AP boundary, which is not surprising considering these discs have developed in the absence of Dpp (*Figure 3E*). Expression of Pent in the dorsal compartment clearly reduced Dlp protein level, and co-localised with Dlp in endosomes (*Figure 3F*). From these experiments we conclude that Pent does not require Dpp or active Dpp signalling in order to internalise glypicans.

## Internalisation of Dally requires dynamin and Rab5 but is clathrin independent

We next wished to address the endocytic mechanism involved in Pent-glypican endocytosis. One of the defining features of glypicans is anchorage to the membrane with a GPI moiety, instead of a conventional transmembrane domain. GPI-anchored proteins are thought to be often internalised in clathrin independent mechanisms which remain poorly understood, particularly in vivo (*Gupta et al., 2009*; *Johannes et al., 2015*). To understand the endocytic mechanism required for Pent mediated endocytosis of glypicans, we combined the Gal4/UAS and LexA/LexO systems to express transgenes in overlapping regions of the disc (*Yagi et al., 2010*). To prevent lethality caused by the inhibition of endocytosis it was necessary to use Gal80ts, with larvae shifted to the restrictive temperature of 30°C 24 hr prior to dissection. In this experimental setup the internalisation and loss of DallyYFP was clearly seen in the region where V5Pent was expressed, and was equivalent in the dorsal and ventral compartments (*Figure 4A*).

One of the most critical proteins for endocytosis is clathrin, which has been described to be required for as much as 95% of endocytic flux (*Bitsikas et al., 2014*). We tested the requirement for clathrin in glypican endocytosis by expressing an RNAi construct against clathrin heavy chain (chc). Staining for clathrin was substantially decreased in the region expressing the chcRNAi, showing that the RNAi is effective (*Figure 4—figure supplement 1A*). However, knockdown of clathrin had no visible effect on the internalisation of Pent or Dally, with many endosomes of Pent and Dally clearly visible in both the dorsal and ventral compartments (*Figure 4B,D*). The immediate destination for many endocytic cargoes is Rab5 positive early endosomes. Knockdown of Rab5 significantly reduced the internalisation of Pent and Dally, with very few endosomes present in the dorsal compartment (*Figure 4C,D*). This suggests that Rab5 is required for endocytic trafficking of Dally internalised by Pent, but clathrin is not.

It has previously been described that clathrin independent endocytosis may require flotillin (*Glebov et al., 2006*). *Drosophila* have two flotillin proteins (also known as Reggie 1 and 2), but Flotillin2 (Flo2) is the most important and is required for Flotillin1 stability (*Hoehne et al., 2005*; *Katanaev et al., 2008*). Knockdown of Flo2 did not significantly reduce Pent or DallyYFP internalisation, suggesting it does not play a role in their endocytosis (*Figures 4D*, *Figure 4—figure supplement 1B*). One of the earliest steps of endocytosis is the scission of nascent endocytic vesicles from the plasma membrane, a mechanism which often requires dynamin. We inhibited dynamin function with a temperature sensitive dominant negative version of Shibire (Shi^TS), the *Drosophila* homologue of dynamin. In order to rule out effects on the secretion of V5Pent, Shi^TS was expressed in the posterior compartment. Shi^TS inhibited internalisation of V5Pent, with fewer endosomes present in the posterior compartment compared to the anterior compartment (*Figure 4—figure supplement 1C,*

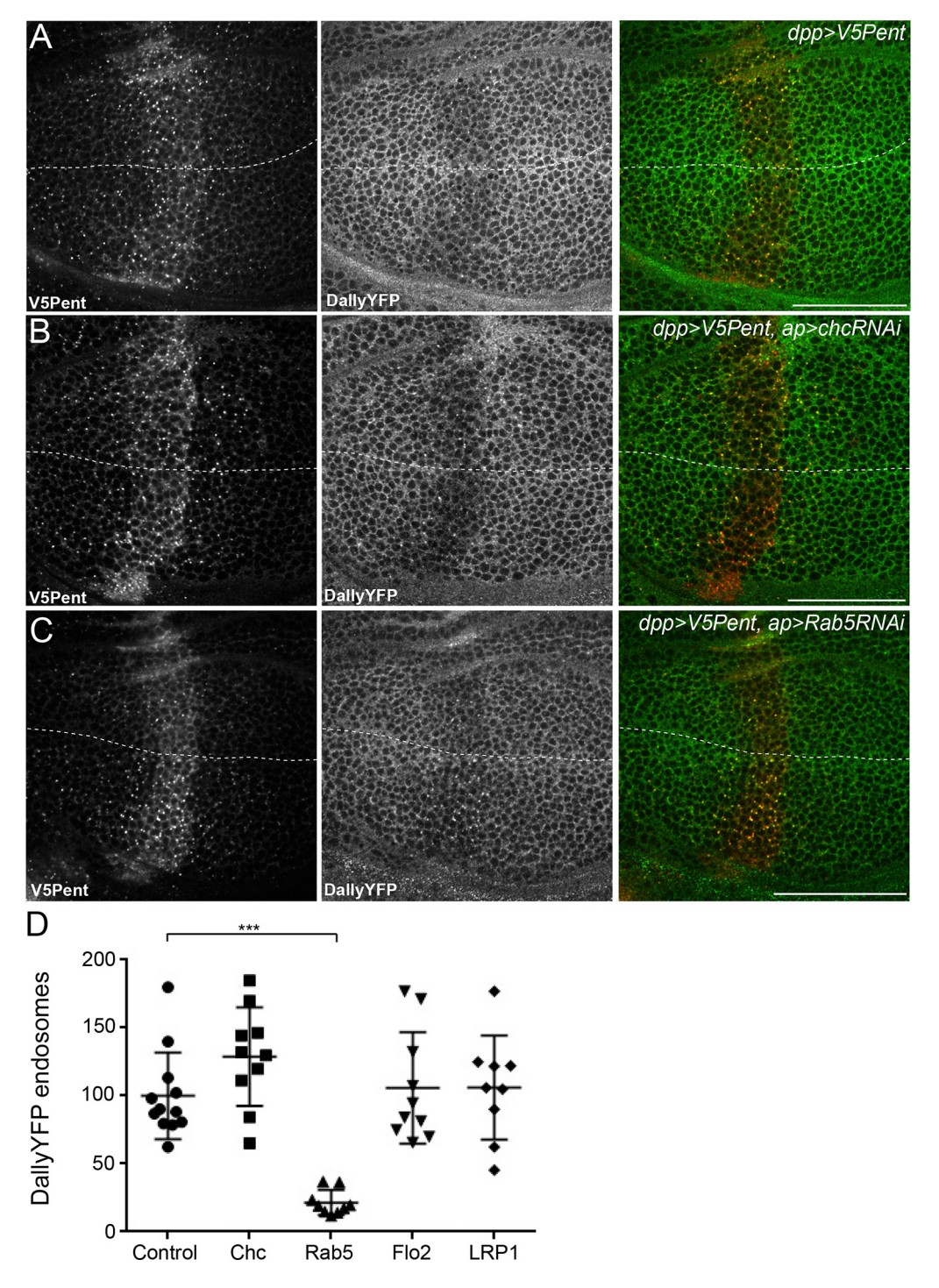

**Figure 4.** The endocytic route of V5Pent and DallyYFP. (A) Control third instar wing disc expressing LexOV5Pent with DppLHV1 LexA driver. Reduction of DallyYFP is seen in both dorsal (above dashed line) and ventral (below dashed line) regions of the V5Pent expressing cells. V5Pent-DallyYFP endosomes are visible in both compartments. (B, C) Wing discs expressing RNAi against Chc (B) or Rab5 (C) with apGal4, restricted to 24 hr with Gal80ts. Internalisation and co-localisation of DallyYFP with V5Pent is reduced with Rab5 but not Chc RNAi. (D) Quantification of DallyYFP endosomes in RNAi experiments. Data is endosomes in dorsal, RNAi expressing compartment divided by control, ventral compartment normalised to control conditions. This shows a clear reduction when Rab5RNAi is expressed, but not with any other RNAi line. (n=9 or greater for each condition,

*Figure 4 continued on next page*

*Figure 4 continued*

***represents p>0.0001, Mann-Whitney U Test). Scale bars are 50 μm in all images. See also *Figure 4—figure supplement 1*.

The following figure supplement is available for figure 4:

**Figure supplement 1.** Endocytosis of Pent and Dally is clathrin and LRP1 independent.

*D*). DallyYFP was seen less frequently in endosomes and was increased at the plasma membrane of posterior compartment cells (*Figure 4—figure supplement 1C–E*). Prolonged expression of Shi^TS caused severe tissue deformation, preventing study of later time points that may have provided a greater inhibition of Dally internalisation. Cumulatively, these data suggest that internalisation of Dally induced by Pent requires Rab5 and dynamin, but is independent of clathrin and flotillin.

## Pent modifies Wg signalling

The data presented so far show that Pent binds to and internalises glypicans. Glypicans function not just in the Dpp signalling pathway, but also the Wingless signalling pathway, among others. We therefore reasoned that if Pent modifies glypican levels in the wing, both Pent loss- and gain-of-function should affect Wg signalling. Indeed, *pent* adult wings occasionally display notches in the wing margin (*Figure 5A*), a phenotype ascribed to a reduction in Wg signalling (*Phillips and Whittle, 1993*). Similarly, there is a clear reduction in the number of chemosensory bristles in *pent* mutant wings, another phenotype which is thought to be due to reduced Wg signalling (*Figure 5B*) (*Couso et al., 1994*; *Kirkpatrick et al., 2006*). These data suggest that Wg signalling is affected in the absence of Pent.

Wingless is distributed in a graded manner, with the highest levels of expression at the edge of the wing pouch and at the DV boundary (*Alexandre et al., 2013*). Loss of glypicans has previously been shown to reduce accumulation of extracellular Wg (*Franch-Marro et al., 2005*; *Han et al., 2005*). As expression of V5Pent reduces glypican levels, we assumed that over-expression of Pent would also affect distribution of extracellular Wg. We chose engal4 to mis-express V5Pent as its expression is confined to the posterior compartment and this crosses the DV boundary from which the highest levels of Wg are produced. In the posterior, Pent expressing region, the peak of extracellular Wg was the same as in the WT control, but the intensity decreased steeper than in the control region (*Figure 5C*). These data are consistent with phenotypes seen in glypican mutant clones (*Han et al., 2005*). We conclude that Pent reduces the ability of cells to maintain Wg on the cell surface, probably through the internalisation of glypicans.

As we have shown that Pent reduces glypican levels, we sought to demonstrate that Pent could modify Wg-phenotypes caused by perturbations of glypican levels. When Dlp was over-expressed with enGal4, adult wings had a scalloped phenotype due to defects in Wg signalling (*Yan et al., 2009*). Absence of Pent enhanced this phenotype, whilst over-expression of Pent completely suppressed the phenotype (*Figure 5D*). Similarly, a well described effect of Dlp over-expression is inhibition of the Wg-target *senseless (sens)* (*Yan et al., 2009*). We found that expression of Dlp in the posterior compartment resulted in no detectable Sens protein, which could be substantially restored by Pent co-expression (*Figure 5—figure supplement 1A–C*). It has recently been shown that one of the functions of glypicans in Wnt signalling is to act as a platform upon which Notum can deacylate Wnt proteins, reducing their signalling potency (*Kakugawa et al., 2015*; *Zhang et al., 2015*). Indeed, reducing the levels of glypicans has been shown to effectively restore defects in wing morphology caused by over-expression of Notum (*Kakugawa et al., 2015*). We reasoned that if our hypothesis that Pent internalises glypicans is correct, then expression of Pent should similarly reduce the effects of Notum over-expression. When Notum was expressed with spaltGal4, substantial regions of the distal adult wing were missing (*Figure 5E*). Co-expression of Pent suppressed this phenotype, restoring wings to a wild type shape, although some vein defects related to Dpp signalling perturbations remained. From these data we conclude that by modulating glypican levels, Pent impacts on Wg signalling activity.

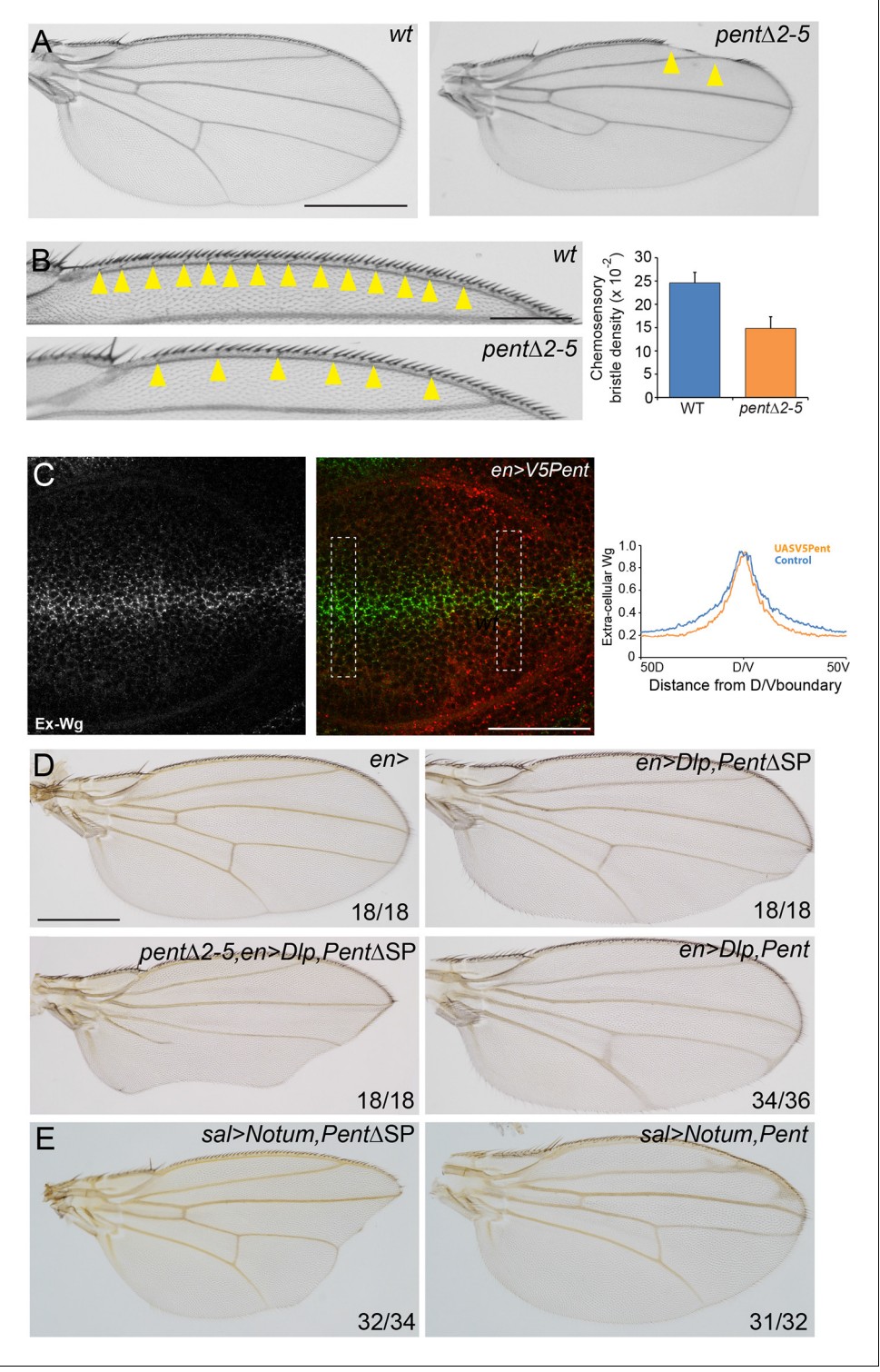

**Figure 5.** Pent modifies Wg signalling. (**A**) Adult wings of WT (left) and *pent△2–5* (right). 9% of *pent△2–5* wings have defects in the wing margin (n=35). Scale bar is 500 µm. (**B**) Adult wings of wild type (upper image) and *pent△2–5* (lower image) flies. Chemosensory bristles are indicated with arrows. The graph shows the density of chemosensory bristles compared to thick, outer mechanosensory bristles in WT and *pent* mutant (n=20). Error bars show standard deviation. (**C**) Third instar wing disc immuno-stained for extracellular Wg and V5Pent. Reduced staining intensity was seen in the posterior compartment which expressed V5Pent. Graph shows quantification of Pent expressing and control areas, as indicated by boxed lines in the merge panel. n=20, scale bar is 50 µm. (**D**)
*Figure 5 continued on next page*

*Figure 5 continued*

Expression of UASDlp-GFP in the posterior compartment caused scalloping of the wing which was more severe in the absence of Pent and suppressed by co-expression of Pent. (E) Expression of Notum with spaltGal4 causes severe wing defects, which are completely suppressed by co-expression of V5Pent. Numbers in (D, E) represent penetrance of displayed phenotypes. See also *Figure 5—figure supplement 1*.

The following figure supplement is available for figure 5:

**Figure supplement 1.** Pent modifies expression of the Wg target gene Sens.

## Discussion

We have previously proposed that the function of Pent in Dpp gradient formation could be to either enhance the ability of Dally to displace Dpp, or to reduce the co-receptor function of Dally (*Vuilleumier et al., 2010*). From the data presented here, we propose that Pent reduces the co-receptor function of glypicans by binding them and inducing their internalisation. We suggest that Pent may promote spreading of Dpp by reducing Dpp co-receptors and therefore Dpp trapping and signal transduction. Internalisation of glypicans is independent of signalling and Dpp itself, which fits the model that removal of glypicans by Pent enhances spreading. Furthermore, we have presented data showing that by regulating glypican levels, Pent is also able to modulate Wingless signalling.

### Pent binds and internalises glypicans

Our work proposes that Pent modulates Dpp signalling via the co-receptors Dally and Dlp. The relative contribution of Dally and Dlp to Dpp signalling is unknown, although both must be removed in order for a reduction in pMad to occur (*Belenkaya et al., 2004*). Data showing that Pent influences the co-receptor but not the receptor itself distinguishes Pent from other BMP signalling modifiers in *D. melanogaster*, such as Crossveinless-2, Short gastrulation and Twisted gastrulation, which bind either the BMP ligand, the receptor, or both (*Ross et al., 2001*; *Serpe et al., 2008*; *Shimmi et al., 2005*). This could reflect the different roles that BMP signalling must fulfil in *D. melanogaster*, where it forms a long-range gradient in the larval wing disc but short-range gradients in the embryo and pupal wing.

Our data show that Pent binds and internalises glypicans. Prior to endocytosis, we believe it probable that glypicans are clustered at the cell surface by Pent, and this might also inhibit their function without necessarily inducing their internalisation. Glypicans share physical properties, notably a GPI anchor and heparan sulphate side chains, upon which Pent binds. We have shown that internalisation of glypicans by Pent requires dynamin and Rab5 but not clathrin. Cell culture experiments have shown that GPI proteins are commonly endocytosed via clathrin independent mechanisms, but this has not been demonstrated before in *Drosophila* (*Johannes et al., 2015*). Many clathrin independent mechanisms have been described in cultured cells, but in vivo evidence for many of them is lacking (*Johannes 2015*). Lipid-rich microdomains, in some cases marked by flotillin, can be involved, but we found no requirement for flotillin in the endocytosis of glypicans by Pent (*Glebov et al., 2006*).

One of the key problems cells must overcome to internalise GPI anchored proteins is that they have no cytoplasmic region to mediate recruitment into endocytic pits. A similar process to that described here, the Hh mediated internalisation of GPC3, is thought to utilise LRP1 in order to communicate with the endocytic machinery (*Capurro et al., 2008*; *2012*). This does not seem to be the case with Pent and Dally, as knockdown of the *Drosophila* homologue of LRP1 does not affect internalisation (*Figure 4—figure supplement 1*). It is possible that protein clustering, and the membrane deformations this has been predicted to cause, may be involved in the internalisation of glypicans by Pent (*Sarasij et al., 2007*). The precise mechanism by which Pent internalises glypicans will be an interesting avenue of future research.

While our data implicate glypicans as the direct target of Pent's activity, we cannot rule out the possibility that the effect on Dpp gradient formation involves the regulation of Tkv levels and/or activity. While Pent does not bind to Tkv directly and Pent over-expression seems not to affect the

levels of membrane bound Tkv, a substantial amount of the receptor is found in Pent- and Dally positive endocytic vesicles. This raises the possibility that Pent might target a specific subpopulation of Tkv for degradation, and that this interaction requires glypicans as adaptors.

## Pent and Dpp signalling

Our data show that Pent binds and internalises Dally and Dlp. Glypicans, in particular Dally, have been shown to regulate the spreading of Dpp, in addition to being essential for Dpp signal transduction itself. The molecular basis for these activities is the binding of Dpp to the heparan sulphate side-chains of glypicans, probably a first step that serves to concentrate Dpp at the surface of the disc epithelium. Glypican-bound Dpp molecules can follow multiple routes, as they can be passed to receptors (promoting signalling), to glypicans of neighbouring cells (promoting ligand dispersion), or can persist on glypicans of the same cell resulting in local ligand enrichment. It is probable that the specific outcome at any position along the morphogen field will depend on the relative levels and activities of the involved factors, i.e. the ligand, receptors and glypicans. A similar balance between glypicans, receptors and ligands has been proposed to explain the biphasic activity of Dlp in Wg signalling in the wing imaginal disc (*Yan et al., 2009*). In the case of Dpp, levels of glypicans need to be tightly regulated to allow for the optimal balance between ligand release, trapping and receptor binding. Our data suggest that Pent contributes to this balance by fine-tuning the levels of glypicans (see *Figure 6* for a model). We propose that in the absence of Pent, glypican levels are too high and this results in excessive ligand trapping and enhanced local signalling. Such local effects would be accompanied with a non-autonomous reduction in ligand spreading and shrinkage of the pMad gradient. An approximation mimicking this situation is artificially elevating levels of Dally in medial regions, which has been shown to locally increase pMad (*Fujise et al., 2003*). We have shown here that this increase in signalling by Dally is at the expense of Dpp spreading to the rest of the disc and the formation of the long range pMad gradient (*Figure 1—figure supplement 1A–F*). This clearly shows that excess Dally can block spreading of Dpp. Notably, *pent* mutants display a similarly compacted activity gradient with high medial and low lateral pMad levels. Importantly, ligand-binding properties of HSPGs have been described to impede ligand spreading in multiple physiological

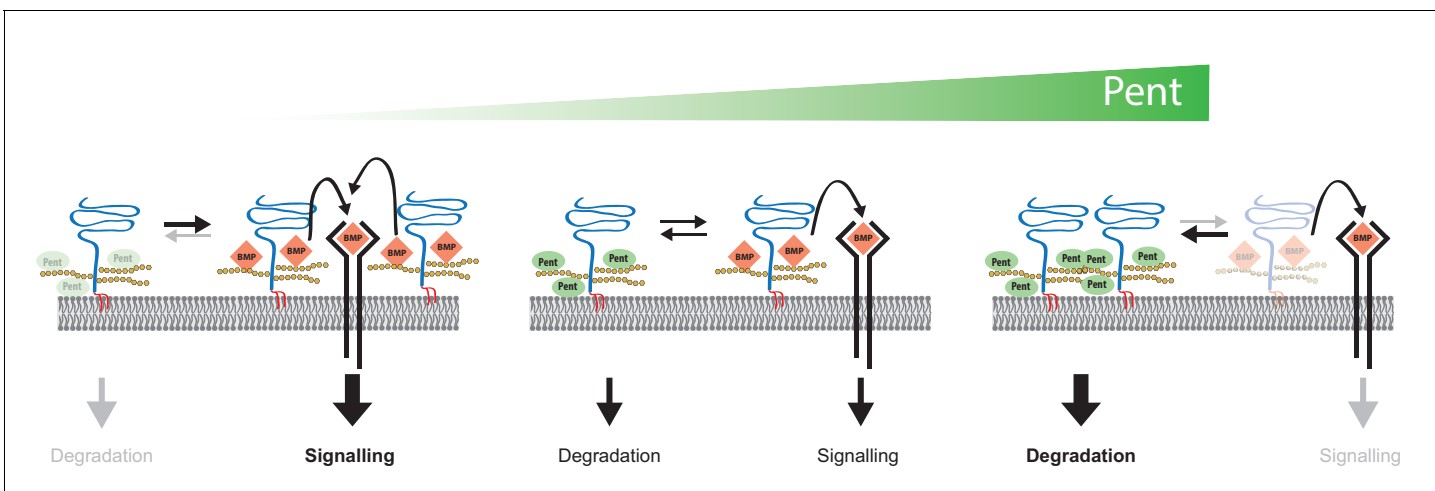

**Figure 6.** Model for Pent´s function in BMP signalling. Pent binds to glypicans and induces their signal-independent internalisation and lysosomal degradation. In the absence of Pent (left), glypican levels on cell surfaces increase. Elevated co-receptor levels enhance local signalling but, at the same time, 'over-trap' Dpp and reduce the pool of ligand that would be available for movement and long-range dispersion. In contrast, excessive levels of Pent (as in our over-expression studies) cause a drastic depletion of surface exposed glypicans (right). As a consequence, and similar to glypican loss-of function conditions, cells fail to bind Dpp and signalling is reduced. Between these two extremes (middle), an optimal concentration of Pent ensures for glypican levels that are high enough to promote signalling but not too high to cause local trapping and 'over-consumption' of Dpp. Thus, in the context of long-range gradient formation, we suggest that adjustable levels of Pent titrate glypicans to ensure for the optimal trade-off between ligands channelled into signalling and ligands available for movement along the morphogen field (see text for details).

contexts, including BMP4 in *Xenopus* early dorso-ventral patterning and FGF10 in its role in branching morphogenesis (*Makarenkova et al., 2009*; *Ohkawara et al., 2002*).

Multiple transcriptional feedback loops are required for the maintenance of the Dpp signalling gradient in the wing. Primary amongst these is the repression of Tkv and Dally transcription by Dpp signalling (*Crickmore and Mann, 2007*; *Fujise et al., 2001*; *Lecuit and Cohen, 1998*; *Tanimoto et al., 2000*). This ensures that receptor and co-receptor levels are low near the Dpp producing cells, allowing Dpp to spread out from the centre of the disc. These feedback loops are important for proper establishment of the Dpp signalling gradient. However, such direct feedbacks targeting the production of molecules with ligand-binding properties may have limitations. In response to a reduction in spreading of Dpp, Tkv and Dally levels would increase to locally compensate the reduction in Dpp signalling activity. Such an increase would, however, further enhance trapping and internalisation of the ligand and, at the level of the whole wing disc, would further block Dpp spreading. From our data we suggest that Pent, a secreted negative regulator of Dpp signalling, fine-tunes the signalling gradient at a different level, by directly adjusting glypican levels and reducing the inbuilt increase in co-receptor and ligand-trapping upon a reduction in the extent of the pMad gradient. This might happen at a critical region of the wing disc, the mediolateral cells, where declining levels of the spreading ligand face increasing levels of the receptor and co-receptor (Tkv and Dally, respectively). Pent, secreted by lateral cells next to this region, could reduce the glypican pool to allow Dpp to overcome excessive ligand trapping and thus promote further spreading. Consistent with such a 'remote' activity, Pent can be detected throughout the wing disc. As Pent is transcriptionally repressed by Dpp signalling and, unlike Tkv and Dally, does not bind Dpp, Pent might be a good candidate for how the system overcomes the inherent limitations of feedback loops involving membrane tethered, Dpp-binding proteins.

## Pent and Wg signalling

The key extracellular signalling molecules of the wing disc, Dpp, Wg and Hh, all bind to glypicans (*Akiyama et al., 2008*; *Franch-Marro et al., 2005*; *Gallet et al., 2008*). The regulatory proteins Pent, Notum and Shifted also bind glypicans, putting glypicans at the centre of signalling regulation in the wing disc (*Bilioni et al., 2013*; *Kakugawa et al., 2015*). Consequently, any factor that affects glypican function, such as Pent, is likely to modify multiple signalling pathways. We have shown that Pent is also able to influence Wg signalling, thus providing a possible link between the Wg and Dpp pathways.

The role of glypicans in Wg signalling is well described and complex. Dlp can stimulate Wg signalling, Wg accumulates on cells over-expressing Dlp and fails to accumulate on cells mutant for Dlp (*Franch-Marro et al., 2005*; *Han et al., 2005*). Similarly, our data show that excess Pent internalises glypicans and reduces extracellular Wg. Precise in vitro assays have shown that low levels of Dlp enhance Wg signalling, but too much Dlp reduces signalling (*Franch-Marro et al., 2005*; *Yan et al., 2009*). Furthermore, recent evidence shows that deacylation of Wg by Notum, which reduces Wg signalling activity, requires glypicans (*Kakugawa et al., 2015*; *Zhang et al., 2015*). It is clear, then, that the level of glypicans must be very finely balanced for Wg signalling to be at the correct level. We propose that the elevated glypican levels observed in the absence of Pent push this fine balance towards inhibition of signalling, due to the increased levels of glypicans sequestering Wg away from the receptor and also increasing the platform upon which Notum can deacylate Wg. Consistent with this conclusion, the effects of Notum and Dlp over-expression can be suppressed by increasing the level of Pent protein.

Interestingly, inactivation of the BMP-response elements in the regulatory region of the *pent* gene locus results in prominent expression of *pent* at the DV boundary (*Vuilleumier et al., 2010*), hinting at an input into *pent* transcription from DV signals. Future studies, including quantitative studies and modelling, should give further insight into pathway interaction and coordination during tissue development by molecules such as Pent.

## Concluding remarks

We propose a model that Pent internalises glypicans to modify multiple signalling pathways. Future work should address the influence of Pent on glypican organisation at the nanoscale, and also the type of membranes at which Tkv and Dally localise, questions that are challenging to answer using

current methods. In order to fully understand the role of Pent in establishment of the long range Dpp gradient, we must first better understand how glycans function in Dpp signalling and how Dpp is spread throughout the tissue.

## Materials and methods

### Immunofluorescence and microscopy

Third instar larvae were dissected and fixed in 4% PFA/Schneider S2 medium. Primary antibodies were incubated overnight at 4°C, and after washing secondary antibodies were incubated at room temperature for 2 hr. The following primary antibodies were used in this study: chicken anti-GFP (Abcam, Cambridge, UK), mouse anti-V5 (Invitrogen), rabbit anti-V5 (Sigma), rabbit anti-Pent (*Vuilleumier et al, 2010*), rabbit anti-Rab7 and Rab11 (*Tanaka and Nakamura, 2008*), rabbit anti-mCherry (Abcam), Guinea Pig anti-Hrs (*Lloyd et al., 2002*), rat anti-Chc (M. Behr, TRM Leipzig), mouse anti-Wg, Ptc, Dlp (Hybridoma Bank, Iowa), mouse anti-ßgal (Promega). Alexa conjugated secondary antibodies were used (Invitrogen). Images were acquired using a Nikon C2 confocal microscope and analysed and adjusted using ImageJ and Adobe Photoshop. All images of larval discs are posterior to the right and dorsal up. Extracellular labelling was performed as described previously for Wg (*Strigini and Cohen, 2000*). For extracellular staining of DallyYFP, an alternative protocol was used. Discs were incubated with anti-GFP nanobody labelled with Abberior AS 635P (Chromotek) for 30 min on ice, and then processed as in *Strigini and Cohen (2000)*. Rab7YFP was used as an intracellular protein control, and showed no staining. Adult wings were dehydrated in 100% isopropanol and mounted in Euparal (Carl Roth GmbH).

### Western blots

Standard conditions were used. 16 discs were loaded in each lane. Antibodies used were rabbit anti-GFP (TP401, Torrey Pines Biolabs), mouse anti-α-tubulin (T5168, Sigma-Aldrich), anti-mouse HRP and anti-rabbit HRP (GE Healthcare).

### Clonal analysis and Gal80ts

FRT mutant clones were generated by heat shocking larvae at 37°C for 1h 72 hr days prior to dissection. LexA and UAS flip out clones were generated by heat-shocking larvae at 37°C for 8 min 72 hr prior to dissection. Crosses were incubated at 25°C, except where indicated that Gal80ts was used. In this case, flies were incubated at 18°C and shifted to 30°C for 24 hr before dissection. For the shibire inhibition experiments, flies expressing UASshi$^{TS}$ were shifted to 30°C for 8 hr before dissection.

### Fluorescence intensity quantification

For co-localisation analyses, at 50 V5Pent endosomes were marked in a Z stack with Z sections 1 μm apart. These marks were then compared to the Dally, Dlp or Tkv channel and scored as positive or negative. If positive, they were then scored as Rab7 positive or negative. The medial region of the dorsal compartment in 10 discs was analysed for each genotype, resulting in a total of over individual 500 endosomes per genotype.

For quantification of the RNAi experiments, Dally endosomes were identified using the 3D objects counter plugin of ImageJ. The dorsal to ventral ratio was calculated by dividing the number of endosomes in the dorsal compartment by the number in the ventral compartment. The mean ratio observed in control was set as 100 and all data was normalised to this value. At least 9 discs were counted for each condition.

For extracellular Wg quantification, maximum projections of the three middle most Z slices were measured in the Pent expressing posterior compartment and in the anterior compartment as control. The measurement was centred on the DV boundary. Twenty discs were measured in total.

To quantify DallyYFP or DlpYFP levels, WT and *pent–2–5* flies homozygous for Dally/DlpYFP were flipped into vials for 6 hr, and dissected 5 days later. All samples were processed together to minimise sample variation. Intensity was measured in a region of interest covering most of the dorsal compartment, centred on the AP border. An average intensity profile of the brightest five slices was measured to reduce noise.

## Fly lines

The DallyYFP (DGRC number 115511) and DlpYFP (DGRC number 115–031) lines were created by the Cambridge Protein Trap Insertion project (*Lowe et al., 2014*). DallymCherry was created using the MiMIC line MIO2220 (*Venken., 2011*). mCherry is inserted between the first two exons, after K90. LexOV5Pent was made by cutting the V5Pent sequence from the UAS construct and ligating into pLOTattB (*Yagi et al., 2010*). UASDallyFlag was made by inserting the Flag tag sequence between R49 and R50. Other lines used were: UASGFPDlp, UASGFPDally, UASGFPDally△HS (S. Eaton), Rab7YFP (*Dunst et al., 2015*), PuntGFP and UASeGFPDpp (M. O'Connor), c40.1gal4 (dppgal4) (M. Hoffman), Rab7YFP (*Dunst et al., 2015*), UASShi$^{TS}$ (BL 5811), pentA17, pent△2–5, UASV5Pent and UAS△SPV5Pent (*Vuilleumier et al., 2010*), UASShfV5 (*Bilioni et al., 2013*), apGal4 (W. Gehring), UAS-VHH-CD8-mCherry (referred to as nanotrap) (M. Affolter), DppLHG$^{TP}$, DppLHV1, DppLG (*Yagi et al., 2010*), dally$^{MH32}$FRT2a, dally$^{MH32}$ dlp$^{20}$ FRT2A/TM3Sb (*Franch-Marro et al., 2005*), Def(3L)ED4421, Def(3L)ED543 and Def(2L)Exel6011 (Bloomington Stock Center), apGal4 gal80ts/cyo (L. Gaffner), Dpp$^{d12}$/cyo (BL2070), dpp$^{14}$ (*Spencer et al., 1982*), Brk$^{XA}$ (*Campbell and Tomlinson, 1999*) EPnotum, FRT40A mad$^{12}$. RNAi lines were from the VDRC collection: chc #103383, rab5 #103495, flo2 #31525, punt #37279, tkv #3059, LRP1 #109605.

## Generation of TkvHA and PentYFP

The PentYFP endogenous fusion protein was generated using previously described methods (*Baena-Lopez et al., 2013*). A new *pent* null mutant was generated by deleting the first exon, which contains the only known transcription start site and the signal peptide, using homologous recombination and inserting an attP site in its place. An integration vector containing: the last 50 bp of the 5'UTR, the first exon with Venus YFP plus CAGTTG inserted after L25, and the first 50 bp of the first intron, was inserted into the attP site.

TkvHA was generated with the same method. The genomic region from 283 bp upstream of the penultimate exon until the stop codon was removed. The same region was re-inserted via attP mediated recombination, with a 3xHA tag inserted directly before the stop codon.

## Acknowledgements

We would like to thank Konrad Basler, Mike O'Connor, Markus Affolter, Marko Brankatschk, Suzanne Eaton, Jean-Paul Vincent, Isabel Guerrero, Hugo Bellen, Akira Nakamura, Matthias Behr, The Developmental Studies Hybridoma Bank at the University of Iowa, Vienna Drosophila Resource Centre and the Bloomington Stock Centre for fly lines and reagents and Janine Seyfferth for excellent technical assistance. This work was supported by the Excellence Initiative of the German Federal and State Governments (EXC294) and a grant from the German Research Council (DFG) to GP (PY72/1-1).

## Additional information

### Funding

| Funder | Grant reference number | Author |
| --- | --- | --- |
| Deutsche Forschungsgemeinschaft | PY72/1-1 | George Pyrowolakis |
| Excellence Initiative of the German Federal and State Governments | EXC294 | Mark Norman<br>Robin Vuilleumier<br>Alexander Springhorn<br>Jennifer Gawlik<br>George Pyrowolakis |

The funders had no role in study design, data collection and interpretation, or the decision to submit the work for publication.

### Author contributions

MN, Designed and performed most of the experiments, Analysis and interpretation of data, Wrote the manuscript; RV, Performed experiments, Conception and design, Acquisition of data, Drafting or revising the article, Contributed unpublished essential data or reagents; AS, Performed experiments,

Conception and design, Analysis and interpretation of data, Drafting or revising the article, Contributed unpublished essential data or reagents; JG, Designed and generated the TkvHA line, Acquisition of data, Analysis and interpretation of data, Drafting or revising the article; GP, Analysed data, Conception and design, Supervised the project, Wrote the manuscript

**Author ORCIDs**

George Pyrowolakis, (iD) http://orcid.org/0000-0002-2142-5943

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
