## [Decision Letter]

Thank you for submitting your work entitled "Pentagone internalises glypicans to fine-tune multiple signalling pathways" for consideration by *eLife*. Your article has been reviewed by three peer reviewers, and the evaluation has been overseen by Hugo Bellen, Reviewing Editor and Janet Rossant, Senior Editor.

The reviewers have discussed the reviews with one another and the Reviewing Editor has drafted this decision to help you prepare a revised submission. While there was considerable interest in the overall concepts in your paper, there were major concerns remaining that need to be addressed experimentally.

All of the issues listed as major will have to be addressed experimentally and discussed much more carefully in light of already published results. The main issues to be addressed focus around

1) looking at pMad distribution

2) looking at Dpp distribution

3) improving on the statistical analyses

An improved Discussion that helps explain apparently inconsistent data would also help the reader.

Summary:

Patterning and growth control of the *Drosophila* wing imaginal disc by Dpp has served as an excellent model system to understand how morphogen gradients control patterning and growth of a developing tissue. The authors previously identified pentagon (*pent*), a secreted feedback factor involved in Dpp morphogen gradient formation. *pent* expression is repressed by *dpp* signalling and Pent protein enhances Dpp morphogen spreading, potentially through binding to dally. However, the mechanism by which Pent regulates the Dpp morphogen gradient shape remained unclear. In this study, the authors find that over-expression of Pent can cause internalization of Dally or Dlp in a Dynamin and Rab5 dependent manner. This effect seems specific to dally and dlp, since Tkv or Punt levels are not influenced by Pent. Based on these results, mainly from over-expression experiments, the authors suggest that Pent-mediated Dally and Dlp internalization and degradation promotes Dpp dispersal and morphogen gradient expansion. As they found that Pent-induced endocytosis of Dally and Dlp the authors expand their analysis to explore a role for Pent in the Wg signaling.

Major issues:

1) The authors do not directly address the effect of Pent-mediated Dally and Dlp internalization on Dpp dispersal and the p-Mad activity gradient. Do the authors observe an expansion of the Dpp gradient or the p-Mad signal in discs over-expressing Pent in the dorsal compartment (for example, dpp-LexA>GFP-Dpp, ap-Gal4>Pent)? Although over-expression of Dally in the stripe of Dpp producing cells at the anterior-posterior border can inhibit Dpp signaling (p-Mad) outside of the stripe, inhibition of Dpp dispersal by Dally in the physiological condition has not been shown previously and is not directly addressed in the current study.

2) The proposed model seems contradictory to the published result, which show that Dally overexpression results in Dpp morphogen gradient expansion by either stabilization of Dpp (Akiyama et al., 2008) or promoting Dpp dispersal (Belenkaya et.al., Cell, 2004). Indeed, it has been shown that overexpression or loss of Dally broadens or narrows the p-Mad gradient respectively, and clones mutant for dally inhibit p-Mad signal behind clones (Belenkaya et al., Cell, 2004), which seems opposite to their interpretation.

3) The authors have previously shown that *pent* mutants and *dally* mutants show similar phenotypes, however double mutants did not enhance or rescue the observed phenotype, suggesting that Pent and Dally act in the same pathway (Vuilleumier et.al., NCB, 2010). If Dally over-expression inhibits Dpp dispersal one would expect that the *pent* mutant phenotype is rescued in pent dally double mutant. In the same paper, the authors showed that Dpp stability is significantly reduced in pent mutants. How can Pent-mediated Dally internalization lead to stabilization of Dpp? These published data including their own seem inconsistent with their interpretation of pent overexpression.

4) It is also difficult to understand the interpretation of the Wg experiment. They conclude that "Pent reduces the ability of cells to maintain Wg on the cell surface, probably through the internalization of glypicans". They interpreted that Dally or Dally-like internalization facilitates Dpp dispersal, in striking contrast, they suggest the opposite for Wg. Is it right that the authors suggest that internalization of glypicans facilitates Dpp dispersal but inhibits Wg dispersal? If so, what might be the reason for this opposing behavior? Furthermore, how does the Wg downstream target gene expression look like in pent mutants or in the discs over-expressing pent in the posterior compartment?

5) Furthermore, it seems contradictory that Pent levels are highest in the lateral region, while according to the mutant phenotype, it exerts its strongest effect in the medial region, reducing medial Dally and Dlp levels. If so, what makes the medial region more susceptible to Pent?

6) The majority of the experiments are done in an over expression setup. Therefore, it is not clear if Pent-mediated Dally internalization plays a critical role during normal development. To address this question, it is important to investigate if up-regulation of Dally or Dlp protein levels in pent mutant are indeed caused by stabilization of these protein, since dally/dlp transcription might be changed in the *pent* mutant. If indeed the protein stability is affected in the *pent* mutant, pulse-chase experiment staining extracellular Dally or Dlp should show longer half-lifes for these proteins. Related to this, what is known about the transcriptional relationship between Dlp and Dpp signaling?

7) The study takes advantage of novel technologies that generated functional YFP fusion proteins and thus permitted in vivo detection of Tkv, Dally and perhaps Dlp (there is some confusion about the Dlp flies utilized- see below). In addition, the authors created one more endogenously tagged line (Dally-mCherry). However, data (or references) demonstrating the normal functionality for these lines is lacking. This is particularly important since the effects of glypicans on signaling vary widely as a function of protein levels, distribution and post-translational modifications.

8) It is not clear if Pent is stimulating Tkv internalization into puncta because the close-ups images shown are in or near the Pent expression domain. If Tkv-Rab7 endosomes are present in the whole disc, then it is simpler to explain why Pent OE does not deplete Tkv, and suggests that the colocalization with Pent (and Dally) is due to later fusion events away from the plasma membrane. A further way to interrogate the uptake of Tkv and Dally+Pent would be to repeat the dependence experiment with clathrin on Tkv puncta formation. (I realize it is hard to see Tkv puncta in many parts of the disc, but is feasible in the medial anterior region, like in Figure 3.) If they have different mechanisms, this would mean they are distinct pathways and may lead to interesting ideas about how Tkv co-receptor interactions influence internalization.

9) Finally, the point that Pent's role is especially critical at the medial/lateral transition zone is interesting, and it may be worth pointing out that this is a parallel way to view its function. The expansion-repression model seems more about scaling, but the description near the end of the discussion frames it as preventing a trapping/sharpening tendency built into the system by the other feedback mechanisms involving Tkv and Dally expression. It may be simple semantics, but both of these angles seem important when thinking about a morphogen gradient.

[Editors' note: further revisions were requested prior to acceptance, as described below.]

Thank you for resubmitting your work entitled "Pentagone internalises glypicans to fine-tune multiple signalling pathways" for further consideration at *eLife*. Your revised article has been evaluated by Janet Rossant (Senior editor), a Reviewing editor (Hugo Bellen), and three reviewers. The manuscript has been improved but there is a remaining issue as outlined below.

One of the key question that the authors should have addressed during this revision was whether Pent-mediated Dally internalization indeed plays a critical role in Dpp dispersal. Unfortunately, the authors could not show convincing data supporting that Dally internalization indeed helps Dpp dispersal under physiological conditions. The data showing that over-expression of Pent leads to internalization of Dally and Dlp is certainly convincing, but this does not mean that such a mechanism plays a role during development.

Without providing such data, the referee and the readers will struggle in understanding the impact of Pent-mediated Dally internalization on Dpp gradient formation. Therefore, it would be helpful and fair to the readers to include the Figure 7 into the main manuscript and discuss it. Furthermore, at several places in the manuscript, the authors state that Pent-mediated Dally internalization modulates BMP signalling (e.g. in the Abstract: "We show that Pent affects BMP signaling through internalization and down-regulation of the Dpp co-receptors […]"); however, the authors do not provide evidence for this and should remove these statements.

It is crucial to clearly point that the evidence provided fully support the interpretation that Pent mediates Dally internalization, but equally clearly state that any connection to morphogen gradient formation is (at this point) speculative. and might not be part of the physiological mechanism of gradient formation.

Another reviewer has the same issue and states: "This revised manuscript is much improved and addresses most of the issues and comments raised before. But there is a general shortcoming perhaps due to the predominant overexpression experimental setting: The reader is left with a good grasp regarding the biological consequences of excess Pent, but with less appreciation on the glypican/Pent balance and its role on signaling when the protein levels are low. Within this frame, it would be more intuitive to assess the signaling outcome as a function of protein concentration and relative level. Such a discussion, or even a schematic representation, could be very helpful in describing the proposed roles for Pent-Glypican in modulation of BMP and Wg signaling and conveying the authors' thoughts on the multiple layers of regulation suggested by their data. "

Please provide the requested schematic diagram and address the textual issues raised.

---

## [Author Response]

Major issues:

1) The authors do not directly address the effect of Pent-mediated Dally and Dlp internalization on Dpp dispersal and the p-Mad activity gradient. Do the authors observe an expansion of the Dpp gradient or the p-Mad signal in discs over-expressing Pent in the dorsal compartment (for example, dpp-LexA>GFP-Dpp, ap-Gal4>Pent)? Although over-expression of Dally in the stripe of Dpp producing cells at the anterior-posterior border can inhibit Dpp signaling (p-Mad) outside of the stripe, inhibition of Dpp dispersal by Dally in the physiological condition has not been shown previously and is not directly addressed in the current study.

We have provided a figure for the reviewers (Figure 7) of the experiment suggested, namely dppLG>GFPDpp, apGal4>V5Pent. Consistent with the removal of glypicans by Pent, we see a drastic reduction in the level of the extracellular Dpp in the dorsal compartment. The assay we use (overexpression of Pent) is intended to address the molecular consequences of Pent binding to glypicans, and results in a very drastic reduction of glypicans from cell-surfaces.

As the reviewer points out, overexpression of Dally in the Dpp stripe causes reduction of pMad activity outside the *dpp*-expression domain (Fujise et al., 2003, Development), which we (and we believe also the authors of the study) interpret as a reduction of ligand dispersal anterior and posterior to the stripe. We have inserted new data (Figure 1—figure supplement 1) which clearly shows this prevents spreading of Dpp itself, and results in a contracted pMad gradient.

Author response image 1.**DOI:**
http://dx.doi.org/10.7554/eLife.13301.018

Given the requirement of Dally as a co-receptor we cannot think of a good experiment for addressing inhibition of Dpp dispersion “in physiological conditions”. Besides the new data included which relies on overexpressing Dally, the only way we think this can be achieved is by alleviating transcriptional repression of Dally by Dpp by manipulations in the genomic locus of dally, an experiment which is certainly beyond the scope of the present study.

Nevertheless, we present and discuss reports where HSPGs have indeed been demonstrated to *block* rather than help dispersal of ligands in other contexts

2) The proposed model seems contradictory to the published result, which show that Dally overexpression results in Dpp morphogen gradient expansion by either stabilization of Dpp (Akiyama et al., 2008) or promoting Dpp dispersal (Belenkaya et.al., Cell, 2004). Indeed, it has been shown that overexpression or loss of Dally broadens or narrows the p-Mad gradient respectively, and clones mutant for dally inhibit p-Mad signal behind clones (Belenkaya et al., Cell, 2004), which seems opposite to their interpretation.

We have discussed the dual role of glypicans above. In short, glypicans are absolutely needed for proper signalling and spreading but since the molecular basis for this is ligand binding, too much of this activity will trap the ligand and shrink the gradient. We have provided clarifications of previous data to show that too much Dally can narrow the pMad gradient, and we have expanded the discussion of these data.

3) The authors have previously shown that pent mutants and dally mutants show similar phenotypes, however double mutants did not enhance or rescue the observed phenotype, suggesting that Pent and Dally act in the same pathway (Vuilleumier et.al., NCB, 2010). If Dally over-expression inhibits Dpp dispersal one would expect that the pent mutant phenotype is rescued in pent dally double mutant. In the same paper, the authors showed that Dpp stability is significantly reduced in pent mutants. How can Pent-mediated Dally internalization lead to stabilization of Dpp? These published data including their own seem inconsistent with their interpretation of pent overexpression.

We understand the confusion of the reviewer, and we have written an extended Discussion to hopefully alleviate this. We believe that due to the multiple roles of Dally, and the fine balancing of the levels required for the correct gradient, it is unrealistic to imagine that removal of Dally completely would restore the *pent* mutant phenotype. In order for this to be true, Dally would have to *only* block spreading, which we do not believe to be the case. Indeed, the correct level of Dally, as determined by transcriptional and protein level feedbacks, is essential for the long range gradient. It is our model that in the absence of Pent, Dally is increased above this optimal level and this prevents the formation of the long range pMad gradient by increasing medial trapping and signalling.

With respect to the stability of Dpp in *pent* mutants, we propose that in the absence of Pent, glypican levels are increased, resulting in an increase in medial signalling. It seems logical that if increased Dally increases signalling, as is believed, and signalling removes Dpp from the cell surface, then increased Dally destabilises Dpp.

4) It is also difficult to understand the interpretation of the Wg experiment. They conclude that "Pent reduces the ability of cells to maintain Wg on the cell surface, probably through the internalization of glypicans". They interpreted that Dally or Dally-like internalization facilitates Dpp dispersal, in striking contrast, they suggest the opposite for Wg. Is it right that the authors suggest that internalization of glypicans facilitates Dpp dispersal but inhibits Wg dispersal? If so, what might be the reason for this opposing behavior? Furthermore, how does the Wg downstream target gene expression look like in pent mutants or in the discs over-expressing pent in the posterior compartment?

With respect to the apparent opposing effect on the spreading of Wg and Dpp, we propose that high levels of Pent internalise glypicans which prevents trapping of either Wg or Dpp in these cells. In the case of Dpp, this would mean an increase in Dpp that can spread further. Recent data has questioned whether Wg really spreads at all, as a non-spreading form of Wg can restore Wg growth (Alexandre et al., 2013). If, as seems possible, Wg mostly signals in an autocrine fashion rather than a long range gradient, then removal of glypicans could not enhance spreading.

As requested by the reviewers, we have provided data showing that the Wg target gene Sens is reduced in *pent* mutants (Figure 4—figure supplement 1). We have also shown that expression of Pent can restore Sens expression which is blocked by Dlp, and have also expanded the discussion of the Wg data.

5) Furthermore, it seems contradictory that Pent levels are highest in the lateral region, while according to the mutant phenotype, it exerts its strongest effect in the medial region, reducing medial Dally and Dlp levels. If so, what makes the medial region more susceptible to Pent?

This is an interesting question, and one we have provided new data to address. Pent is highest laterally as it is repressed by Dpp signalling in order to form the negative feedback loop previously described. Our immunostaining could preferentially label intracellular Pent, which is likely highest laterally, over surface bound Pent. To address this directly, we expressed a membrane tethered antiGFP nanotrap in the Dpp source in a disc expressing PentYFP under endogenous regulation. In this scenario, we see very high levels of PentYFP at the centre of the disc, stabilised on the nanotrap. This very clearly shows that Pent is found all over the disc, not just laterally.

Also, as Dpp is produced in the centre and moves outwards, in order to influence the spreading of Dpp Pent must exert its effect away from the lateral regions. Finally, recent data has shown that the lateral regions may not require Dpp for growth (Harmansa 2015), so it is not inconsistent that Pent exerts its effect away from the lateral region.

6) The majority of the experiments are done in an over expression setup. Therefore, it is not clear if Pent-mediated Dally internalization plays a critical role during normal development. To address this question, it is important to investigate if up-regulation of Dally or Dlp protein levels in pent mutant are indeed caused by stabilization of these protein, since dally/dlp transcription might be changed in the pent mutant. If indeed the protein stability is affected in the pent mutant, pulse-chase experiment staining extracellular Dally or Dlp should show longer half-lifse for these proteins. Related to this, what is known about the transcriptional relationship between Dlp and Dpp signaling?

The secreted nature of Pent makes it impossible to study its effects in mutant clones, which would be invaluable for addressing the loss of function phenotype. Therefore we have used over-expression to evaluate the molecular function of Pent.

The point about protein stability and pulse chase is a very valid one, and something we have tried to address many times. In our hands, conventional extracellular staining does not work for glypicans as it massively favours the basal region of the disc. This is why we describe extracellular labelling using a nanobody in the paper, which penetrates the tissue much better. We tried many pulse chase experiments using this extracellular labelling approach, but we could not obtain consistent data. The primary problem is that labelled extracellular DallyYFP does not significantly decrease over the time window we used, which was up to 8h chase. This is perhaps not surprising as previous pulse chase experiments have shown that in wild type discs Dpp is stable after 3h, and we believe it likely that Dally is more stable than Dpp. We found that the integrity of the discs suffered when we chased for longer than 8h. Further methods are needed to address this question better, but this was not possible in the time window available. We have provided some data of our pulse chase attempts in a figure for the reviewers (Figure 8)

Author response image 2.**DOI:**
http://dx.doi.org/10.7554/eLife.13301.019

Despite the lack of consistent data from our efforts to follow the stability of glypicans in pulse-chase experiments, we now provide additional evidence that both glypicans are increased in pent mutant discs by directly monitoring their levels in Western blots, as suggested by the reviewers (see also below).

The regulation of Dlp by Dpp signalling has not been previously described, and unlike Dally its expression profile does not suggest it is regulated by Dpp signalling. Nevertheless, we directly test for this and provide direct evidence that shows that Dlp is not a target of Dpp signalling (Figure 2—figure supplement 1).

7) The study takes advantage of novel technologies that generated functional YFP fusion proteins and thus permitted in vivo detection of Tkv, Dally and perhaps Dlp (there is some confusion about the Dlp flies utilized- see below). In addition, the authors created one more endogenously tagged line (Dally-mCherry). However, data (or references) demonstrating the normal functionality for these lines is lacking. This is particularly important since the effects of glypicans on signaling vary widely as a function of protein levels, distribution and post-translational modifications.

This is a very valid point, and we have provided adult wing analyses and pMad staining for all of the transgenic lines used in the study (Figure 1—figure supplement 2). We recently became aware of some problems with the TkvYFP line, which, although homozygous viable, produces wings with mild venation defects. We have replaced all experiments conducted with this line with a new genomic engineered Tkv version. The line expresses C-terminally tagged Tkv (Tkv-HA) from the endogenous locus and was tested to be fully functional. This data is also provided in Figure 1—figure supplement 2.

8) It is not clear if Pent is stimulating Tkv internalization into puncta because the close-ups images shown are in or near the Pent expression domain. If Tkv-Rab7 endosomes are present in the whole disc, then it is simpler to explain why Pent OE does not deplete Tkv, and suggests that the colocalization with Pent (and Dally) is due to later fusion events away from the plasma membrane. A further way to interrogate the uptake of Tkv and Dally+Pent would be to repeat the dependence experiment with clathrin on Tkv puncta formation. (I realize it is hard to see Tkv puncta in many parts of the disc, but is feasible in the medial anterior region, like in Figure 3.) If they have different mechanisms, this would mean they are distinct pathways and may lead to interesting ideas about how Tkv co-receptor interactions influence internalization.

These are both very interesting points. We have added to the manuscript that the presence of Dally and Tkv in endosomes could be due to endosome fusion. We attempted the experiment suggested by the reviewer, which is the expression of V5Pent in a disc with TkvHA and also knocking down clathrin by RNAi. This required many crosses, and sadly we could not get data of a publishable standard. We believe that this is an interesting but minor point, and does not directly affect the strength of the conclusions.

*9) Finally, the point that Pent's role is especially critical at the medial/lateral transition zone is interesting, and it may be worth pointing out that this is a parallel way to view its function. The expansion-repression model seems more about scaling, but the description near the end of the discussion frames it as preventing a trapping/sharpening tendency built into the system by the other feedback mechanisms involving Tkv and Dally expression. It may be simple semantics, but both of these angles seem important when thinking about a morphogen gradient.*

This is an interesting point. We have included some of this in the Discussion, but as we do not address scaling experimentally we believe it would be unnecessarily confusing to address this in the Discussion.

In addition, to these comments, and in accordance with the editor´s suggestions, we have quantified the internalisation of Dally, Dlp and Tkv by Pent, and the frequency of their co-localisation with the late endosome marker Rab7. We have also quantified the impact of the RNAi experiments, and this has greatly clarified this figure.

[Editors' note: further revisions were requested prior to acceptance, as described below.]

*One of the key question that the authors should have addressed during this revision was whether Pent-mediated Dally internalization indeed plays a critical role in Dpp dispersal. Unfortunately, the authors could not show convincing data supporting that Dally internalization indeed helps Dpp dispersal under physiological conditions. The data showing that over-expression of Pent leads to internalization of Dally and Dlp is certainly convincing, but this does not mean that such a mechanism plays a role during development.*

[…]

*Please provide the requested schematic diagram and address the textual issues raised.*

We agree and have tuned-down our statements when it comes to the role of Pent in gradient formation. In the current version we have been very careful to clarify in all relevant parts (Abstract, Discussion) that the suggested role of Pent (which is based on our findings on the effects of the protein on glypican levels) is indeed a working model, which is, however, consistent with and takes into account all phenotypic data.

We have also included and discuss “Figure 7” in the manuscript as suggested (current Figure 2—figure supplement 2) which was also expanded to show effects on the activity gradient (pMad staining).

We also agree that it is a good idea to discuss our findings in the light of the activity/concentration balances. We have followed the suggestion of the second referee and provide a schematic that recapitulates our data and model (Figure 6). In order to conform to the first part of the reviewer´s comments and suggestions, the model focuses on consequences of alterations of relative protein levels in local signaling. We have deliberately avoided including consequences on global morphogen gradient formation in this schematic as they are considered by the reviewers premature and speculative. We have restricted our thoughts regarding the role of Pent and its function as a feedback regulator in the text of the Discussion emphasizing that this is a working model.